# Mitochondrial Analysis of Sparidae Species to Detect a New DNA Barcoding Marker for *Dentex gibbosus* to Utilize against Fraud

**DOI:** 10.3390/foods12183441

**Published:** 2023-09-15

**Authors:** Iolanda Venuti, Marina Ceruso, Tiziana Muscariello, Rosa Luisa Ambrosio, Angela Di Pinto, Tiziana Pepe

**Affiliations:** 1Department of Veterinary Medicine and Animal Production, University of Naples Federico II, Via F. Delpino, n. 1, 80137 Naples, Italy; iolandavenuti@gmail.com (I.V.); tizianamuscariello@gmail.com (T.M.); rosaluisa.ambrosio@unina.it (R.L.A.); tiziana.pepe@unina.it (T.P.); 2Department of Veterinary Medicine, University of Bari Aldo Moro, Prov. le Casamassima, Km 3, Valenzano, 70010 Bari, Italy; angela.dipinto@uniba.it

**Keywords:** fish species authentication, mtDNA, mitogenomics, *Dentex gibbosus*, Sparidae

## Abstract

*Dentex gibbosus* (Pink dentex) is a fish species of increasing economic interest in the Mediterranean Sea that is consumed both whole and processed. The growing value of this sparid in European markets is responsible for its substitution with fraudulent species. The distinctive morphologic feature of *D. gibbosus* is the conspicuous hump on the forehead in the older and larger specimens. However, the head is regularly convex in young individuals, requiring high skills and competencies for correct identification. Authentication becomes even more challenging in the case of prepared and processed products. Therefore, the molecular characterization of Pink dentex plays a crucial role in preventing commercial fraud with species substitution. This paper proposes a comparative mitogenome analysis between 19 sparid species of commercial interest as a tool to accurately design species-specific primers targeting a fragment of the *NAD2* gene for the identification of *D. gibbosus*. We successfully detected Pink dentex DNA both using endpoint and real-time PCR. The findings showed the high specificity of the designed primers, demonstrating this a suitable, fast, and cost-effective method that could be used for the unambiguous identification of Pink dentex. This innovative approach for sparid authentication is expected to contribute to seafood traceability, public health assurance, integrity, and the credibility of the seafood industry.

## 1. Introduction

Pink dentex (*Dentex gibbosus*, Rafinesque 1810) stands out as one of the most economically interesting species within the Sparidae family. It is geographically distributed along the West African coast from Portugal to Angola [1]. *Dentex gibbosus* also inhabits the Mediterranean Sea, excluding the northwestern coastal regions and the northern Adriatic Sea [2], and it currently holds a classification of “Least Concern” in the Red List of Threatened Species in the Mediterranean Sea [3]. The catches of Pink dentex are documented by the Food and Agriculture Organization (FAO) only in the Mediterranean, notably in Croatia and Turkey. Precisely, Croatia’s fishing yield was recorded at two metric tons, while Turkey exhibited a noteworthy aquaculture production of 61 tons (FAOFishStatPlus, 2018). This sparid economic and organoleptic importance is responsible for its deep value in European markets and, consequently, for its fraudulent substitution with species of less economic value [4].

*D. gibbosus* can be found on the market in the form of a whole specimen or as prepared and processed products [5]. A distinctive morphologic feature of *D. gibbosus* is the head profile. Specifically, the older and larger specimens develop a conspicuous hump (gibbosus in Latin) on the front. However, in young individuals, the forehead is regularly convex, requiring high skills and competencies for correct identification, even when its morphologic features are not modified by processing [6]. In prepared and processed products, species identification requires laboratory investigations. In this context, the molecular characterization of Pink dentex plays a crucial role in preventing commercial fraud regarding seafood on the market. Several mitochondrial (mt) genes, such as cytochrome b-*Cytb*, cytochrome c oxidase I-*COI*, *16S*, and *12S*, are presently and indiscriminately used for the identification of all fish species in prepared and processed products [7,8,9,10]. However, standard mitochondrial DNA (mtDNA) markers demonstrate proficiency for fish species discrimination with a high degree of variation in nucleotide sequences, yet it exhibits reduced discrimination when the nucleotide resemblance between species is notably elevated [11,12]. A Sparid mtDNA alignment revealed about 80% [13] more homology than other fish species [14]. This outcome shows that Sparidae translated sequences are very analogous to each other. Research on the authentication of Sparidae species in seafood has been going on for many years [13,15,16,17]; it shows that a comparative study of the mtDNA is an effective approach for identifying new and species-specific barcoding markers. In the previously mentioned research [15,16,17], we identified two new molecular markers for sparids. The first one is an *NAD5* gene fragment, useful for all sparid species authentication, and requiring PCR amplification and Sanger sequencing for correct species detection. The second is an *NAD2* gene marker for *Pagellus erythrinus* and *Dentex dentex* direct detection, and not requiring sequencing due to its degree of genetic divergence among sparids higher than standard genetic markers. Currently, this research group has sequenced and deposited in GenBank the complete mitochondrion of numerous sparid species [18,19,20,21,22,23,24,25]. The objective of this work was to compare and analyze the updated mtDNA of Sparidae species currently present in Genbank to find a new molecular tag useful for the unequivocal barcoding of *D. gibbosus* by designing species-specific primers targeting a fragment of the *NAD2* gene. The proposed method is simple and rapid, and requires inexpensive lab equipment, giving important support to competent national authorities responsible for monitoring and for deterring dishonest market chain actors from fraudulent seafood labeling.

## 2. Materials and Methods

### 2.1. mtDNA Genome Data and Sample Collection

The complete mtDNA of the species *D. gibbosus* was compared with other Sparidae complete mitogenome sequences available in GenBank (Table 1).

The *D. gibbosus* specimens were sampled considering different geographical origins, as shown in Table 2. According to EC Reg. 1224/2009, fishing companies provided geographical coordinates for Pink dentex samples. All the specimens were from the 37 FAO area since it contains most of the species in circulation (www.aquamaps.org, accessed on 21 July 2023). The sampling areas followed the GSA (Geographical Sub-Area) partition by GFCM (General Fisheries Commission for the Mediterranean) (Resolution GFCM/31/2007/2). Pink dentex samples were used to assess the species-specificity of the PCR primers. The evaluation was extended to 28 other fish species, as shown in Table 3. The fish species other than *Dentex gibbosus* were carefully chosen to include (i) those used for substitution (e.g., other *Dentex* species), (ii) the most phylogenetically correlated sparid species (e.g., *Dentex dentex*) (Figure 1), and (iii) commercially important Mediterranean fish species.

All the fish species considered in this study (Table 2 and Table 3) were directly frozen on board after fishing at −20 °C and immediately transported in insulated containers to the Food Inspection Laboratory at the Department of Veterinary Medicine and Animal Production (University of Naples Federico II). The classification at the species level was carried out according to their anatomical and morphological characteristics. The categorization at the species level was conducted based on their anatomical and morphological attributes.

### 2.2. Genomic DNA Extraction

Genomic DNA (gDNA) extraction and DNA quantification of the extracted gDNA were performed as previously described [16,17]. DNA concentration was adjusted to 50 ng/µL and purity A260/A280 ratio within a range of 1.8–2.0 was considered. DNA integrity was verified by electrophoretic analysis in 1% agarose gels.

### 2.3. mtDNA Comparative Analysis

The study of the complete mtDNA of the nineteen sparids was carried out using different bioinformatics tools to identify the most efficient genetic marker for Pink dentex detection and authentication.

The evolution of Sparidae species history was deduced by using the Maximum Likelihood method and the Tamura-Nei model [36,37]. The tree with the highest log likelihood (−133,865.97) is presented (Figure 1). Initial tree(s) for the heuristic search were automatically acquired by applying Neighbor-Join and BioNJ algorithms to a matrix of pairwise distances computed using the Tamura-Nei model. Subsequently, the topology with the utmost logarithmic likelihood value was chosen. This analysis encompassed the complete mitochondrial nucleotide sequences of 19 sparid species. The ultimate dataset comprised a sum of 23,259 positions. Evolutionary analyses were conducted in MEGA11 [38]. Hamming distance algorithm [39], overall mean *p*-genetic distance, and pairwise and multiple alignments on DNA sequences were determined to assess the genetic divergence among genes and species as previously described [13,16,17]. The selected reference mitogenome for comparison was *Dentex gibbosus*, ac. number MG_653593 [21]. Nucleotide sequence gene-by-gene variability was determined using the analytical approach of MEGA 6.0. Variable site evaluation is shown with the total site number (Variable sites/Total # of sites) after the exclusion of missing/gap sites from all the nucleotide and amino acid sequences. The gene-by-gene p-genetic distance among *D. gibbosus* and *D. dentex* was performed using the Maximum Composite Likelihood model [36,37].

### 2.4. NAD2 Primer Design and Specificity Test

*NAD2* species-specific primers were identified by eye after multiple alignments of the sparids’ mtDNA sequences using BioEdit Sequence Alignment Editor [40]. Multiple Primer Analyzer was employed to confirm melting temperature (Tm), secondary structure, self-annealing, and inter-primer binding (Thermo Fisher Scientific, Waltham, MA, USA). The specificity of *NAD2* primers for *D. gibbosus* identification was computationally assessed using the Unipro UGENE software [41] comparing results with fish species shown in Table 1 and Table 3 for which NAD2 sequences are available in databases. In addition, *D. gibbosus NAD2* gene was blasted in Genbank to find the first 100 species most similar in nucleotide sequence.

### 2.5. End-Point and Real-Time PCR

PCR reactions were carried out on total gDNA from 10 fresh specimens of *D. gibbosus* (Table 2) and 28 other fish species (Table 3). To verify the intra-species variability of the NAD2 amplified fragment, all *D. gibbosus NAD2* amplicons were subjected to sequencing and genetic intraspecific variation was analyzed. PCR amplifications were performed as previously described [13]. The annealing temperature was at 63° (primer set No 1) and 65 °C (primer set No 2) for 30 s, and extension at 72 °C for 45 s. COI primers [42] were utilized as a control. PCR products were examined through electrophoretic analysis on a 1.5% agarose gel and observed using the Universal Hood II Gel Doc System (Bio-Rad, Hercules, CA, USA) to assess the presence of fragments of the expected length. Amplicons were purified using the QIAquick PCR Purification Kit (Qiagen, Hilden, Germany) and sequenced by the Sanger method with the Automated Capillary Electrophoresis Sequencer 3730 DNA Analyzer (Applied Biosystems, Foster City, CA, USA) by Bio-Fab research s.r.l. (Rome). *NAD2* sequences were studied using the BioEdit Sequence Alignment Editor [40]. All the gained sequences were compared with those available in GenBank using BLAST analysis to assess the concordance between morphological and molecular analyses [43].

The DNA extracted was used as template for Real-Time PCR (qPCR) analysis carried out in the StepOne Real-Time PCR System (Applied Biosystem) at 1:10 dilutions, with the species-specific primers (Table 4).

PCR volume of each sample was 20 μL, with 10 μL of 2X Optimum qPCR Master Mix with SYBR Green (GeneSpin, Milano, Italy), 3, 6, or 9 pmol/μL for each primer and 5 μL of diluted DNA template. The following thermal program was applied: 95 °C for 5 min, followed by 40 cycles at 95 °C for 10 s, annealing at the appropriate melting temperature (Tm) and time for each primer pair, and 60 °C for 30 s. The melt-curve analysis was performed at temperatures ranging from 65 to 95 °C with a ramping rate of 0.5°/5 s. COI primers [42] were used as an internal control. Reproducibility, robustness, and sensitivity were assessed as previously described [16,17]. The results shown (Figure 6) are with 3 pmol/µL of each primer.

## 3. Results

### 3.1. Sparidae mtDNA Comparative Analysis

The sparid cladogram showed that, in agreement with previous work [44,45,46], *D. gibbosus* and *D. dentex* are the phylogenetically closest sparid species (Figure 1).

This result was confirmed by hamming genetic distance, showing that *D. gibbosus* and *D. dentex* have the lowest genetic dissimilarity (11%) based on the complete mitochondrial genome sequence (Figure 2).

In order to find the most useful molecular tag to unequivocally barcode Pink dentex, a p-distance analysis was conducted gene-by-gene among *D. gibbosus* and all sparids. The results about genetic dissimilarity were the following: *ATP6* (20%), *ATP8* (17%), *COI* (14%), *COII* (13%), *COIII* (14%), *Cytb* (15%), *NAD1* (17%), *NAD2* (19%), *NAD3* (17%), *NAD4* (19%), *NAD4l* (16%), *NAD5* (17%), *NAD6* (18%). These findings suggested that the highest dissimilarity value was obtained for *ATP6* (20%), *NAD2* (19%), and *NAD4* (19%) (Figure 3).

These results were confirmed by a gene-by-gene nucleotide and amino acid sequence variability evaluation among Sparidae species (Table 5) that showed that the genetic divergence among genes has the highest values for the genes *NAD2* and *ATP6* (50% and 53% of nucleotide sequence variability, respectively).

To find the best molecular marker for the identification of Pink dentex, the gene-by-gene p-genetic distance among *D. gibbosus* and *D. dentex* was calculated since these are the phylogenetically closest sparid species. The results showed that the *NAD2* gene has the highest value of genetic dissimilarity in percent (Figure 4).

The gene-by-gene p-distance analysis among *D. gibbosus* and *D. dentex* highlighted that the *NAD2* gene displays more sequence distance than all the other genes (Figure 5), showing the highest divergence value (11%). Results are in accordance with previous studies on *P. erythrinus* and *D. dentex* [16,17] and confirmed that the *NAD2* gene has the best suitability for obtaining an unequivocal genomic *D. gibbosus* barcode.

### 3.2. NAD2 Amplification and Analysis

Based on the previous results of the complete mtDNA analysis of Sparidae species, *D. gibbosus* species-specific primers were designed for the amplification of a 290 bp fragment of the *NAD2* gene. As reported in Table 5, two primer sets (1 and 2 forward and reverse) were selected and tested. Primer set No. 2 is composed of shorter primers that could be useful for processed products.

As previously described (Section 2), the species specificity of the designed primers (290 bp) for *D. gibbosus* was first tested in silico, comparing their sequence with NAD2 genes of the fish species shown in Table 1 and with the first 100 species more genetically similar from GenBank. The results confirmed the species specificity of the primers for *D. gibbosus*. PCR results then confirmed that both the primer sets allow amplification in 10 *D. gibbosus* specimens with different geographical origins (Figure 5).

A PCR was also carried out on 28 different fish species (Table 3) to verify the primer specificity. As expected, the amplification was detected only in Pink dentex (data not shown).

To verify the intra-species variability of the *NAD2* amplified fragment, all *D. gibbosus NAD2* amplified fragments were subjected to sequencing and the genetic intraspecific variation was analyzed. Results ranged from 0% to 0.5%, with two different nucleotides found in two out of ten specimens (Dg1 and Dg2 from the Aegean Sea, Appendix A). This result further supported the *NAD2* reliability for species characterization and confirmed the data obtained for *D. dentex* [16]. The sequences were blasted using international databases and the results allowed us to confirm the exact species identification for *D. gibbosus*, with similarity scores ranging between 98% and 100%.

### 3.3. Real-Time PCR

A Real-Time PCR (qPCR) is a quick and effective approach for performing official controls. The presence/absence of a target species can be verified without the electrophoresis step, and the method does not require highly skilled operators. We tested both our species-specific primers using a SYBR Green qPCR to verify if this method is appropriate for *D. gibbosus* identification. A specific amplification in all the *D. gibbosus* analyzed specimens (Ct between 11 and 20 and a Tm of the products of about 85.4 °C) was observed. All the other tested species resulted in an undetectable signal level, indicated by a Ct of 37, due to a very low amount of non-specific product with a Tm different from that expected (80/81 °C) (Figure 6).

## 4. Discussion

The current Regulation (EU) 1379/2013 on the common organization of the markets in fishery and aquaculture products has established detailed rules for the accurate identification and labeling of seafood products in the national market. According to the Italian official list of seafood trade names (annex I of ministerial decree n. 19105 of September the 22nd, 2017), the only species that can be sold under the name of “Dentice” is *D. dentex.* Therefore, *D. gibbosus* and all the other species belonging to the same genus have to be specifically clarified, being declared as “Dentice gibboso”. However, non-compliance with labeling still remains, especially between phylogenetically related species [47].

Within the Sparidae family, and in particular for *D. gibbosus*, different species substitution scenarios can occur: (i) sparids can be replaced by species belonging to a different family (e.g., Sparidae vs. Luthianidae and Lethrinidae); (ii) fraud may regard similar species of the Sparidae family belonging to a different genus (e. g. *Dentex* spp. vs. *Pagrus* spp.); (iii) fraudulent replacement can concern specimens belonging to the same genus of the family (e. g. *Dentex dentex* vs. *Dentex gibbosus*).

Regarding the type of replacement among species of distinct families (i), the genus Dentex, Pagellus, or Pagrus, being pink-red in the color of the livery, are susceptible to fraudulent substitution with some species from the Lutianidae family (*Lutjanus bohar*, *L. sebae*, *L. malabaricus*) native to the Indian and Pacific Ocean [48]. The replacement can also occur between sparids with a silver-gray coloration of the livery (snappers and porgy) and members of the Letrinidae family (*Lethrinus atlanticus*) [48]. In this case, an examination of the dental configuration becomes imperative, given that the Sparidae, Luthianidae, and Lethrinidae families share the subsequent traits: ventral fins positioned on the thorax, a single dorsal fin on the dorsal margin, the ovoidal body shape, and caudal bilobate fin [49,50].

With regards to fraud among species belonging to a different genus (ii), it is possible to distinguish *Dentex* species (e.g., *Dentex gibbosus*, *D. macrophthalmus*, and *D. angolensis*) from Pagellus, Pagrus, Diplodus, and Spondyliosoma species through a dental table examination, whereas to effectively recognize the species belonging to the genera Boops or Sarpa, the differences in their teeth are imperceptible, thus it is necessary to use the microscope.

Finally, it is possible to differentiate *D. gibbosus* from *D. dentex* since the teeth are thinner, even if less delicate [48,51]. *D. dentex* is different from the other species belonging to the same genera because it has caniniform teeth that are much more robust and of a bigger diameter compared to those quite linear of *D. barnardi* and *D. canariensis*. *D. macrophthalmus* has protruding and slightly arched caniforms, which may be confused with *D. gibbosus.* The identification of the adult individuals of Pink dentex is simpler since they present a marked protuberance on their forehead, but this characteristic is not evaluable in juveniles [52].

Given these marked morphological similarities and the high degree of operators’ specialization needed, the application of molecular techniques has become essential for distinguishing fish as prepared or processed products. DNA barcoding has provided a powerful tool for fish species authentication [53]. *COI* and *CYTB* have been demonstrated to be effective barcodes in species identification [54]. However, the lack of adequate polymorphism to discriminate closely related species based on genetic distance still represents a limitation [55]. A PCR using species-specific primers is a reliable alternative for the identification of closely related species. A species-specific PCR appears to be highly suitable for species authentication purposes [55]. A 290 bp fragment might not be amplifiable in highly processed products, but it should be noted that the Pink dentex fraudulent substitution regards, in particular, products like fillets [4]. In this paper, we presented a simple and efficient method that can be quickly executed for identifying *D. gibbosus*. The assay can easily be applied for routine and high-throughput analyses using conventional or real-time PCR. As reported in previous studies [16,17], the *NAD2* gene presents a high degree of genetic divergence, covering the entire nucleotide sequence. This characteristic allowed the design of species-specific markers for *D. gibbosus* and direct identification, bypassing the sequencing step. The development of species-specific primers for Pink dentex will be useful as a rapid screening tool for fraud identification, contributing to the enforcement of fisheries regulations.

## 5. Conclusions

The mislabeling and replacement of valuable fish species with others of lesser quality are some of the major concerns in the seafood supply chain. Substitution fraud may be responsible for the lack of appreciation of *Dentex gibbosus* on the market. Currently, consumers focus their preferences on “medium-value” fish species (e.g., cod, sea bass, etc.), causing the overfishing of a small number of species. The molecular characterization of fish species of growing economic importance, like the Pink dentex, supports their diffusion for the sustainable management of fisheries. The design of species-specific DNA barcoding markers represents a reliable, specific, and rapid means for unambiguous fish species authentication. Our research findings are expected to provide a significant perspective for the detection of commercial fraud in seafood, contributing to the molecular traceability of fishery products in agreement with Regulation (EU) 1379/2013 (European Commission, Brussels, Belgium, 2013) and achieving a healthy development of a blue economy through a sustainable approach to marine ecosystems and biodiversity.

## Figures and Tables

**Figure 1 foods-12-03441-f001:**
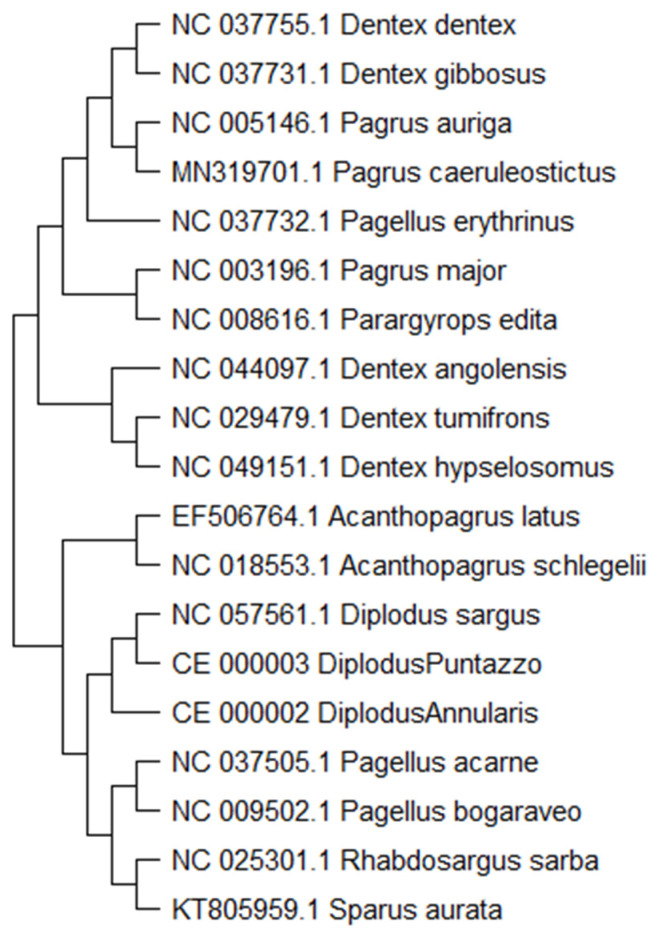
Sparidae species, cladogram.

**Figure 2 foods-12-03441-f002:**
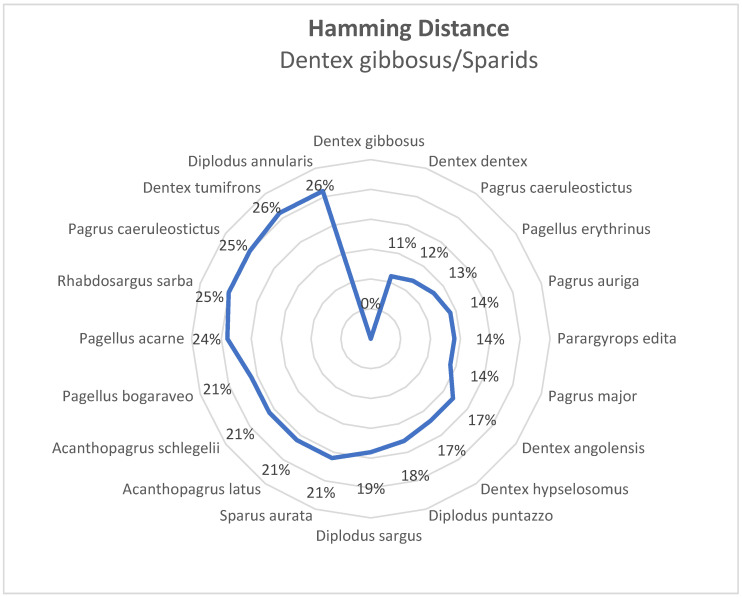
Hamming dissimilarity in percent between *D. gibbosus* and the other sparids evaluated in this research. The order was set up disposing the species from the more similar to the more distant from *D. gibbosus*, counterclockwise.

**Figure 3 foods-12-03441-f003:**
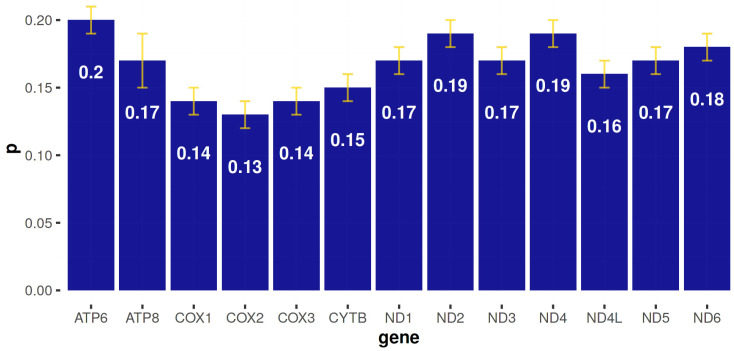
Gene-by-gene p-distance (±SD) analysis among *D. gibbosus* and all sparids.

**Figure 4 foods-12-03441-f004:**
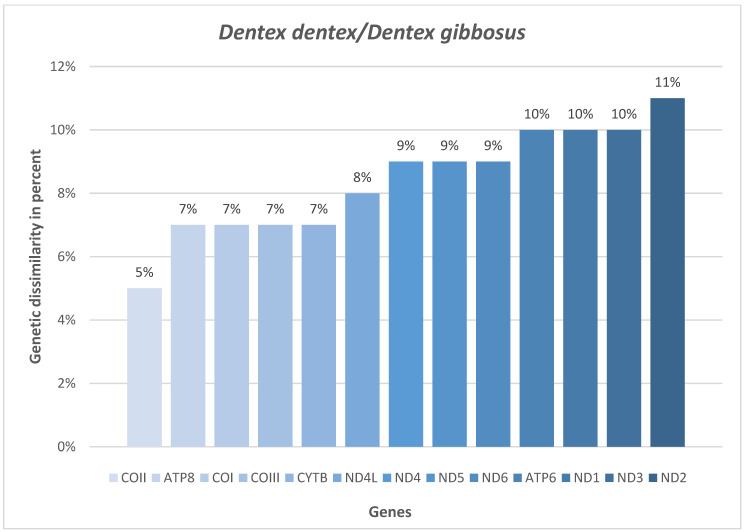
Gene-by-gene p-distance analysis in percent among *D. gibbosus* and *D. dentex*.

**Figure 5 foods-12-03441-f005:**
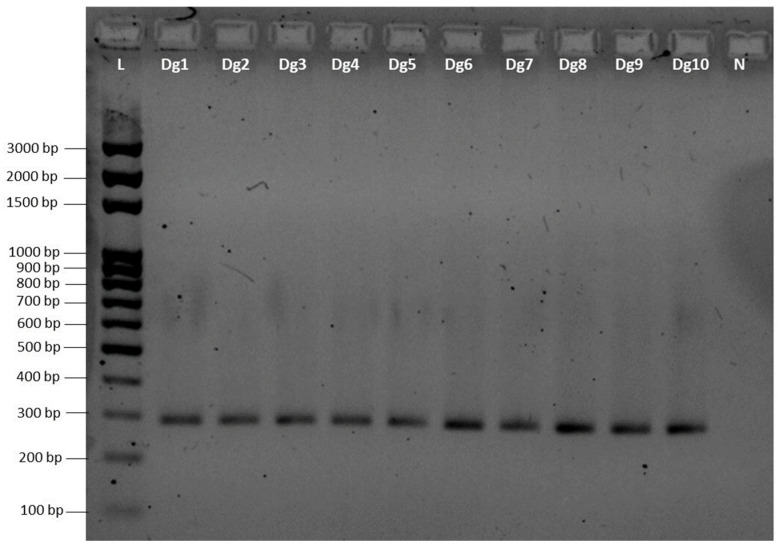
Gel electrophoretic image. End-point PCR amplification of the *NAD2* fragment in ten geographically disjunct specimens of *D. gibbosus* (from Dg1 to Dg10) using primer set No.1. The same results were obtained using primer set No. 2. Abbreviations as in Table 2. N: negative control; L: 100 bp ladder.

**Figure 6 foods-12-03441-f006:**
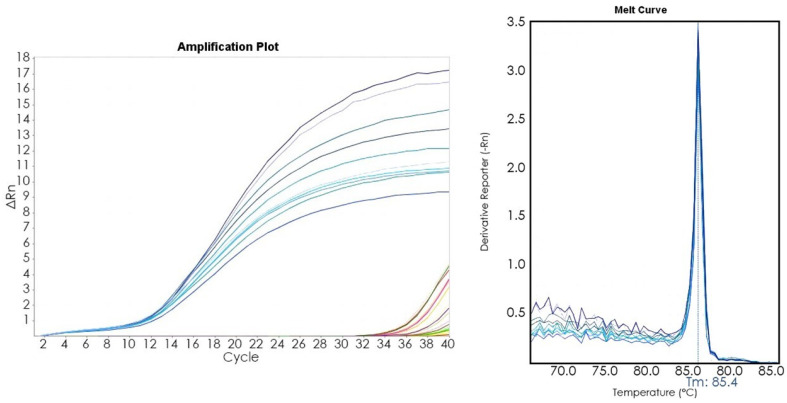
SYBR Green real-time PCR. On the left: amplification plot of Dg1-Dg10 (in blue) and the species reported in Table 1 (in different colors). On the right: melting temperature (Tm) of Dg1-Dg10 (in blue). A single peak at 85.4 °C with *D. gibbosus* DNA as a template indicates the specificity of the designed primers. Specific results were obtained both for primer sets No. 1 and No. 2.

**Table 1 foods-12-03441-t001:** Complete mtDNAs compared in this research.

No	Sparid Species	Ac. Number	FAO Fishing Areas	References
1	*A. latus*	NC_010977	FAO 71	[26]
2	*A. schlegelii*	JQ_746035	FAO 71	[27]
3	*D. angolensis*	NC_044097	FAO 47	[22]
4	*D. dentex*	MG_727892	FAO 37	[19]
5	*D. gibbosus*	MG_653593	FAO 37	[21]
6	*D. tumifrons*	NC_029479	FAO 71	[28]
7	*D. cervinus*	NC_064339	FAO 37	[25]
8	*D. hypselosomus*	NC_049151	FAO 61	[29]
9	*D. puntazzo*	MT319027	FAO 37	[23]
10	*D. sargus*	NC_057561	FAO 37	[24]
11	*P. acarne*	MG_736083	FAO 37	[20]
12	*P. bogaraveo*	NC_009502	FAO 27	[30]
13	*P. erythrinus*	MG_653592	FAO 37	[18]
14	*P. auriga*	AB_124801	FAO 37	[31]
15	*P. caeruleostictus*	MN319701	FAO 34	[32]
16	*P. major*	NC_003196	Andalusia (Spain) fish market	[33]
17	*P. edita*	EF_107158	FAO 71	[26]
18	*R. sarba*	KM_272585	Guangdong (China)	[34]
19	*S. aurata*	LK_022698	Jaffa (Israel) fish market	[35]

**Table 2 foods-12-03441-t002:** *D. gibbosus* samples origin: GSA and geographic coordinates.

Sample Abbreviation	GSA	Latitude	Longitude
Dg1	22—Aegean Sea	37.600.221	24.800.000
Dg2	22—Aegean Sea	38.167.163	25.594.178
Dg3	6—Northern Spain	41.865.047	3.746.190
Dg4	4—Algeria	37.352.717	1.778.274
Dg5	17—Northern Adriatic Sea	45.110.439	12.900.912
Dg6	10—Southern and Central Tyrrhenian Sea	39.403.227	13.096.640
Dg7	18—Southern Adriatic Sea	40.988.637	18.420.498
Dg8	19—Western Ionian Sea	37.115.696	16.486.263
Dg9	9—Ligurian and North Tyrrhenian Sea	42.083.042	10.191.733
Dg10	26—Southern Levant Sea	31.097.963	28.197.500

**Table 3 foods-12-03441-t003:** Fish species other than *Dentex gibbosus* considered in this research. Common names are from the ASFIS List of Species for Fishery Statistics Purposes (http://www.fao.org/fishery/collection/asfis/en, accessed on 14 July 2023). The “category” column specifies the principal reason for including the species in this study on *D. gibbosus.*

No	Scientific Name	Family	Common Name	Abbreviation	Category
1	*Arnoglossus laterna*	Bothidae	Mediterranean scaldfish	Al	Commercially important Mediterranean fish species
2	*Aulopus filamentosus*	Aulopidae	Royal flagfin	Af	Commercially important Mediterranean fish species
3	*Boops boops*	Sparidae	Bogue	Bb	Commercially important Mediterranean fish species
4	*Cepola macrophthalma*	Cepolidae	Red bandfish	Cm	Commercially important Mediterranean fish species
5	*Cheimerius nufar*	Sparidae	Santer seabream	Cn	Commercially important Mediterranean fish species
6	*Chelidonichthys lucerna*	Triglidae	Tub gurnard	Cl	Commercially important Mediterranean fish species
7	*Coris julis*	Labridae	Rainbow wrasse	Cj	Commercially important Mediterranean fish species
8	*Dentex angolensis*	Sparidae	Angolan dentex	Dn	Used for substitution
9	*Dentex dentex*	Sparidae	Common dentex	Dd	Used for substitution and phylogenetically related
10	*Dentex tumifrons*	Sparidae	Yellowback seabream	Dt	Used for substitution
11	*Diplodus annularis*	Sparidae	Annular seabream	Da	Commercially important Mediterranean fish species
12	*Diplodus sargus*	Sparidae	White seabream	Ds	Commercially important Mediterranean fish species
13	*Lithognathus mormyrus*	Sparidae	Sand steenbras	Lm	Commercially important Mediterranean fish species
14	*Lophius piscatorius*	Lophiidae	Angler (=Monk)	Lp	Commercially important Mediterranean fish species
15	*Mullus barbatus*	Mullidae	Red mullet	Mb	Commercially important Mediterranean fish species
16	*Pagellus acarne*	Sparidae	Axillary seabream	Pa	Commercially important Mediterranean fish species
17	*Pagellus erythrinus*	Sparidae	Common pandora	Pe	Phylogenetically related
18	*Pleuronectes platessa*	Pleuronectidae	European plaice	Pp	Commercially important Mediterranean fish species
19	*Sebastes capensis*	Sebastidae	Cape redfish	Sc	Commercially important Mediterranean fish species
20	*Scophthalmus maximus*	Scophthalmidae	Turbot	Sm	Commercially important Mediterranean fish species
21	*Solea solea*	Soleidae	Common sole	Ss	Commercially important Mediterranean fish species
22	*Spondyliosoma cantharus*	Sparidae	Black seabream	Spc	Commercially important Mediterranean fish species
23	*Thunnus thynnus*	Scombridae	Atlantic bluefin tuna	Tth	Commercially important Mediterranean fish species
24	*Thunnus albacares*	Scombridae	Yellowfin tuna	Ta	Commercially important Mediterranean fish species
25	*Thunnus obesus*	Scombridae	Bigeye tuna	To	Commercially important Mediterranean fish species
26	*Trachurus trachurus*	Carangidae	Atlantic horse mackerel	Tt	Commercially important Mediterranean fish species
27	*Trisopterus minutus*	Gadidae	Poor cod	Tm	Commercially important Mediterranean fish species
28	*Xyrichtys novacula*	Labridae	Pearly razorfish	Xn	Commercially important Mediterranean fish species

**Table 4 foods-12-03441-t004:** Species-specific *NAD2* primers for *D. gibbosus.*

Name	Sequence (5′–3′)	Tm °C	CG%	nt	A	T	C	G
1_F_Gib	gcttcttctagccctaggaattacatcaacc	70.2	45.2	31	8.0	9.0	10.0	4.0
1_R_Gib	ctttgatatgaagccggtgagtgggggc	78.1	57.1	28	5.0	7.0	4.0	12.0
2_F_Gib	gcttcttctagccctaggaattac	61.8	45.8	24	5.0	8.0	7.0	4.0
2_R_Gib	ctttgatatgaagccggtgagtg	66.8	47.8	23	5.0	7.0	3.0	8.0

**Table 5 foods-12-03441-t005:** Gene-by-gene nucleotide and amino acid sequence variability evaluation among Sparidae species (Table 1).

Sequence Variability (Variable Sites/Total Sites)				
Genes	Nucleotide		Amino Acid	
NAD1	389/975	40%	41/325	13%
NAD2	530/1047	50%	122/340	35%
COI	500/1566	32%	33/521	6%
COII	223/691	32%	22/230	10%
COIII	258/786	33%	31/262	12%
ATP8	72/165	42%	18/55	32%
ATP6	308/684	53%	57/228	38%
NAD3	136/351	39%	16/117	14%
NAD4L	116/297	39%	14/99	14%
NAD4	610/1386	43%	94/462	20%
NAD5	756/1839	41%	127/613	21%
NAD6	234/522	45%	48/174	6%
CYTB	406/1141	36%	34/381	9%

## Data Availability

Data is contained within the article or Appendix A.

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
