# Peer review of "Mitochondrial Analysis of Sparidae Species to Detect a New DNA Barcoding Marker for Dentex gibbosus to Utilize against Fraud"

_foods, 2023, doi:10.3390/foods12183441_

Round 1

Reviewer 1 Report

Reviewer Comments to Author:

Reviewer: 1

Recommendation: Should be in Foods after major revision.

Comments:

The manuscript described a PCR method for Dentex gibbosus identification. Authors firstly made a comparative mitogenome analysis between 19 sparid species of commercial interest. Then, they designed species-specific primers targeting a fragment of the NAD2 gene for the identification of D. gibbosus. They declare that the proposed method is specific, suitable, fast, and cost-effective , and could be used for the unambiguous identification of Pink dentex. However, the developed assay was not adequately validated and lack of novity.

Major issues:

1. What is the novelty of this work? PCR was developed in 1983 and it has been widely used for nucleic acid amplification and detection. Authors used PCR to detect a sequence, I think this is quite normal and lack of novelty.

2. In this manuscript, authors verified the detection specificity by computer analysis and one PCR plot (Figure 6). They only showed the melt curve of 10 D. gibbosus but not show the melt curve of other species. Authors should further verify PCR results by sequencing.

3. Authors find this special sequence by different gene analysis. Why not directly blast the whole sequence of different fish to find the special sequence of Dentex gibbosus?

4. Authors should provide the detailed sequence of 290bp and relative blast result of different fishes.

5. The quality of figure 6 is low. Figure 5 should label the size of each ladder.

The English showed be comprehensively improved. 

Reviewer 2 Report

The manuscript introduces a novel approach that addresses a critical need in the field of D. gibbosus identification. The research findings have the potential to drive significant advancements, impacting the industry's future practices and strategies. The clear and concise presentation of the research, along with the easily understandable scientific discussions, ensures that the manuscript is not only informative but also engaging for a diverse readership.

The research delves into the concern of mislabeling and substituting valuable fish species within the seafood supply chain, with a specific focus on Dentex gibbosus (Pink dentex). The study's primary objective is the development of molecular techniques for precise species authentication, aimed at countering fraudulent practices prevalent in the market. The subjectaddresses the pivotal challenge of species mislabeling and substitution within the seafood sector. This work's intent was to compare and analyze the updated mtDNA of sparidae species accessible in Genbank, with the purpose of identifying a new molecular marker for unambiguous barcoding of D. gibbosus. The endeavor revolves around designing species-specific primers targeting a fragment of the NAD2 gene. The methodology undertaken is comprehensive and robust. However, detailing the validation process of the species-specific markers could enhance its clarity. For greater solidity, a more extensive sample size and a broader range of processing levels in validation, along with comparative analyses involving alternative methods, are worth considering.

I have a few constructive suggestions that, if implemented, could further enhance the overall quality of this commendable work:

 1.      In the introduction section, I recommend using the full form of "Food and Agriculture Organization (FAO)" initially before employing the abbreviation "FAO" in subsequent references. This approach will cater to a wide-ranging audience.

 2.      In Tables 1 and 3, it is advisable to replace the unusual superscript "N° 'o' with the more common abbreviation "No" for numbering.

 3.      There appear to be alignment issues with some of the tables. Ensuring consistent alignment will improve the visual presentation.

 4.      To enhance the clarity of Figure 3, I suggest removing the grids from the background. This adjustment will make the figure easier to interpret.

 5.      Figure 6 could benefit from an improved resolution for enhanced visual clarity. Removing the grids and increasing the font size of labels would greatly contribute to this.

Throughout the manuscript, minor grammatical corrections are needed. I encourage the authors to review and address these linguistic matters.

Round 2

Reviewer 1 Report

The authors have addressed my comments expect the sequencing suggestion. I think it is quite important to verify the PCR amplification results by sequencing. I support publication after minor revision.

The English should be polished.

Author Response

Dear Reviewer,

please see the attached modified paper, with deep editing of English language.
